# A Model for Consumer Acceptance of Insect-Based Dog Foods among Adult UK Dog Owners

**DOI:** 10.3390/ani14071021

**Published:** 2024-03-27

**Authors:** Joanne Pinney, Montserrat Costa-Font

**Affiliations:** 1Royal (Dick) School of Veterinary Studies, University of Edinburgh, Easter Bush Campus, Midlothian EH25 9RG, UK; 2Rural Economy, Environment & Society Research Group, Scotland’s Rural College (SRUC), King’s Buildings, West Mains Road, Edinburgh EH9 3JG, UK; montse.costafont@sruc.ac.uk

**Keywords:** insect-based dog foods, pet food, dog owners, consumer acceptance, alternative protein

## Abstract

**Simple Summary:**

The use of alternative and insect proteins in pet feed is becoming more common. However, little research has been conducted to date in respect of the drivers for consumers’ acceptance of insect-based dog foods. This study aimed to investigate the factors influencing consumer attitudes towards insect-based dog foods and the decision to try and buy such products. Consumer acceptance of insect-based dog food is multi-faceted, including social, cultural and ethical components. Social norms had the strongest influence on consumer attitudes. Consumer food preferences for animal welfare, health and environment; attitudes towards uses of animals; beliefs about insect sentience; disgust; and perceptions of benefits and risks also had significant influence on attitudes towards insect-based dog food. In order to allow dog owners to make informed decisions in line with their ethical preferences, further research is needed to establish the overall health and welfare implications of insect-based dog foods on the animals involved in production, as well as the companion animals, who are the ultimate consumers.

**Abstract:**

The use of alternative proteins is becoming more common in pet feed, and insect-based dog foods (IBDFs) are becoming more widely available. However, little research has been conducted to date in respect of the drivers for consumers’ acceptance of IBDF. This study aimed to investigate the acceptance of IBDF among adult UK dog owners and the factors influencing the decision to try and buy such products. A theoretical model was developed following a review of the existing literature. An online survey of 280 participants was carried out and the results were analysed using structural equation modelling (SEM) to test the theoretical model. The following constructs all had a significant impact on attitudes towards IBDF and/or intentions to try and buy IBDF: food preferences for animal welfare, health and environment; attitudes towards uses of animals; beliefs about insect sentience; disgust; perceptions of benefits and risks; and social norms. Social norms had the strongest influence of any single construct. Consumer acceptance of IBDF is multi-faceted including social, cultural and ethical components, and it is likely that the better availability of information and opportunities for consumers to familiarise themselves with IBDF would help to drive consumer acceptance. In order to allow dog owners to make informed decisions in line with their ethical preferences, further research is needed to establish the overall health and welfare implications of IBDF on the animals involved in production, as well as the companion animals, who are the ultimate consumers.

## 1. Introduction

Insects are increasingly promoted as a more environmentally friendly and sustainable alternative protein source in the diets of humans and non-human animals [1]. Edible insect sectors are emerging in countries not traditionally associated with the consumption of insects [2]. Previous research has revealed the complexity of European attitudes towards insects as food and feed [3]. In the West, it is thought that insects may have a greater potential as animal feed than human food due to cultural biases [4]. Insect-based feed has potential as an alternative to soybean meal and fishmeal for meat-production animals [5] and is also available for companion animals [6], including dogs.

Traditionally, pet-food production has been largely based on the use of by-products from human food production and, as such, has held a unique role in helping to reduce waste in food production [7]. However, changing consumer demands towards the humanisation of pets alongside an increased quantity and quality of meat products within pet foods means that many commercial pet foods provide nutrients in excess of physiological needs, use ingredients fit for human consumption, and/or are overfed to pets, resulting in wastage and obesity in pets [7,8,9]. This has a significant effect on the environmental impact of companion dogs and cats. The global environmental footprint of pet food has been calculated as being equivalent to around twice the land area of the UK, with greenhouse gas emissions from pet food putting the pet food industry at the level equivalent to emissions from countries such as Mozambique or the Philippines [10]. Insects are increasingly viewed as an important part of future sustainability in human and animal food systems as they can be produced using less land and water compared to traditional livestock and with lower greenhouse gas emissions [11].

In the UK, the most common sources of protein in pet foods are: meat and meat meal; fish and fishmeal; dairy products and eggs; vegetables and legumes; and cereals [12]. Seven species of insect are currently authorized for use in pet food in the EU and the UK, of which the most commonly used are house crickets; yellow mealworms; and black soldier flies [12].

To date, very little research has been carried out as to consumer attitudes towards using insects as a food source for companion animals. Studies have suggested a certain level of acceptance and positive attitudes towards insects as animal feed for farmed animals [13] and fish [14]. However, it is not clear whether there would be a similar level of acceptance towards insect-based feed for pets.

The few studies which have explored consumer attitudes to insect-based dog foods (IBDFs) [15,16] have found that perceptions of the disgustingness and the benefits of insect foods predicted both people’s willingness to eat these foods themselves and their willingness to feed them to their dogs. However, consumers may be less keen on insect-based feed for their pets than for poultry and fish [16].

There are a number of niche IBDF brands on the market, and more well-known brands have slowly started to add insect-based lines to their ranges. As insect-based diets for dogs become more readily available and promoted as an environmentally friendly alternative to diets based on traditional meat sources, more analysis is needed in respect of the attitudes towards such diets in the humans who may purchase them. Greater consumer acceptance of insect-based dog foods could help to drive further exploration and innovation in respect of alternative proteins in companion animal diets. If such products become more mainstream, manufacturers and retailers are more likely to realise the commercial benefits from economies of scale and reduced waste and energy in production, which could help to reduce the prices of pet-food products with a lower environmental impact.

The present research aims to investigate consumers’ acceptance of IBDF among adult UK dog owners and the factors influencing consumer choice to try and buy such products. It will assess how and to what extent certain consumer attitudes, perceptions and beliefs, as identified below, influence attitudes towards IBDF and consumer willingness to try and buy these products.

## 2. Literature Review

### 2.1. Food Neophobia and Disgust

Food neophobia is characterised in humans as a reluctance to eat and/or an avoidance of novel or unfamiliar foods [17]. Most people in European societies rarely experience insects as a food source [18] and this lack of familiarity means that, for many, insect-based foods would be considered novel.

Food neophobia has been found to be negatively correlated with the willingness to eat insects [19,20]. Verbeke [16] concluded that food neophobia makes the largest contribution to consumers’ readiness to adopt insect-based foods as a substitute for other meat. Other studies have also identified food neophobia as a significant factor in consumers’ acceptance of insect-based food and feed [21,22,23].

Previous findings also suggest that the strongest determinant of consumers’ acceptance of insects as animal feed was a person’s own willingness to eat insect products [16]. Therefore, it is likely that food neophobia on the part of the consumer would influence attitudes towards IBDF and their willingness to try such products. The intention to try is a strong predictor of the behaviour of eating insects, and people with lower food neophobia scores have been found to be more likely to try and, consequently, to eat insects [19].

Whilst feelings of disgust towards a novel food can be linked to food neophobia, both food neophobia and disgust can make independent contributions to the intention to eat insects, with indications from some research that disgust has a considerably higher overall impact on attitudes [24]. Disgust, whether in respect of the sight or proximity of insects or the thought of eating them, has been identified as a significant factor in negatively influencing Western consumer attitudes towards eating insects [22,25,26,27] and a main barrier towards the consumption of insects [28].

Higa et al. [15] found that disgust in respect of insect consumption was a significant predictor of people’s willingness to eat insect-based foods themselves and their willingness to feed them to their dogs.

### 2.2. Perceptions of the Risks and Benefits of Insect-Based Food

Trust in the safety of novel or unfamiliar food products is not easy to establish and it is believed that disgust in respect of insects is driven by a fear of contamination and disease [29].

Studies have recorded mixed results in respect of the impact of perceptions of risk on consumer choices related to direct and indirect entomophagy and insect-based food products. Szendrő et al. [30] found that perception of risk (or lack thereof) played the most important role in consumers’ acceptance of a product originating from an animal fed with insect meal. Other studies have also demonstrated a correlation between risk perception and attitudes to entomophagy [31]. However, pathogen disgust and perceived infectability did not consistently predict insect-eating disgust, willingness to eat insects or actual insect-tasting behaviour in Jensen et al.’s [29] research.

In addition, risk perception did not sequentially mediate the proposed relationships between regulator trust and purchase intention in Legendre et al.’s [32] study and, instead, relationships among constructs were mediated by perceptions of benefits. Verbeke [16] found that whilst insect-based feed for animals was perceived to have a lower microbiological safety and to be at risk of the presence of off-flavours and allergens, benefit perceptions were stronger and outweighed risk perceptions as a determinant of accepting the use of insects in animal feed. Ruby et al. [27] found that the perception of risk was not a significant predictor of an overall willingness to eat insects, but perceptions of benefits were. Menozzi et al. [22] also found that belief in the benefits of eating an insect-based food product significantly affected attitudes and intentions to try such products. It is likely that the balance of perceived risks and benefits of insect consumption will have an impact on the attitudes of consumers towards IBDF.

### 2.3. Attitudes towards Uses of Animals, Beliefs Regarding Insect Sentience and Animal-Welfare Food Preference

There is a distinct lack of research in respect of the impact of attitudes towards uses of animals and their influence on attitudes towards the consumption of insects. However, concern for animals, measured using the Animal Attitude Scale [33,34] has been demonstrated to influence dietary choices such as meat-eating and vegetarian and flexitarian diets [35]. It would therefore be expected that higher levels of concern for animals would correlate with the choice to follow a diet that includes less meat or alternatives to traditional meat products. Concern and empathy for animals measured using an adapted Animal Attitude Scale has also been associated with a preference for and intent to purchase products produced with higher standards of animal welfare [36].

However, the relationship between attitudes towards animals and attitudes towards insect consumption is likely to be more complex as speciesism and disgust for insects may contribute to a disparity between consumers’ general attitudes towards animals and attitudes towards insect species. This study will, therefore, also explore beliefs regarding insect sentience separately from attitudes towards the use of animals.

There is evidence to suggest that various species of insects are capable of a range of cognitive abilities [37]. There is significant debate over whether or not insects can experience pain and suffering [38], with some research concluding that insects are capable of experiencing emotional capabilities [39,40] and even “play” behaviours [41]. However, insects are regarded by many as lacking sentience and as being beyond moral consideration [2], and very little is currently known about the capacity of insects to suffer in farming systems [37]. Many people find the moral issues surrounding farming and killing insects less compelling than issues raised in relation to more commonly consumed animals [27].

Few studies have explored the relationship between consumers’ moral views regarding insect sentience, and/or their ability to feel pain or experience suffering, and the willingness to eat insects or feed them to other animals. In studies involving participants from the USA and India, perceptions of insect suffering were not found to be significant predictors of an overall willingness to eat insects [27], and attitudes in respect of the morality of killing insects did not significantly predict acceptance [26].

Attitudes towards insects are likely to be complex. For example, it has been speculated that for some vegetarians, e.g., those primarily motivated by environmental reasons, insects may be an acceptable animal protein source [27]. For others with strong views on animal rights, the consumption of insects may be no more acceptable than the killing of more traditionally consumed species. However, this is yet to be explored.

Concern and empathy for animals have been associated with a preference for and intent to purchase products produced with higher animal-welfare standards [36]. Studies in the United States [42] and Germany [43] found that pet owners were interested in farm animal welfare and willing to pay for welfare-friendly pet food. The main factors contributing to a preference for animal welfare in food products are socio-demographics, ethics and attitudes, product characteristics and public roles [44]. Animal-welfare concerns and desires to minimise animal suffering have been reported as important motives in the willingness to change eating behaviours and move to plant-based diets [45,46]. Interventions appealing to animal-welfare concerns have also had an effect on reducing meat consumption and purchase intentions, at least in the short term [47].

There has been very little research into the influence of animal-welfare food preference on consumers’ willingness to try or buy insect-based food products and limited research into its influence on consumers’ willingness to try and buy non-plant-based meat alternatives. Some studies have linked perceptions of animal-welfare benefits to more positive attitudes towards lab-grown meat [48,49,50]; studies have suggested that consumers are unlikely to adopt novel meat substitutes for animal-welfare reasons alone and this preference has a lower priority for consumers than other factors [51,52].

Insect-based food products are also promoted as a more sustainable alternative to traditional meat-based products, and Mazzocchi et al. [53] identified a strong link between environmentally conscious consumers and those with values linked to animal wellbeing.

### 2.4. Environmental and Healthiness Food Preference

The perception of positive environmental effects was found to be the most important outcome of eating products containing insect flour for participants in Menozzi et al.’s [22] study. In addition, further research has shown that the importance of the environmental impact of food choices positively influenced the perceived suitability of both insect-based and plant-based meat substitutes [21]. Verbeke [16] also found that insect-based feed was perceived to be much more sustainable compared to food products obtained from animals fed on conventional diets. These studies included participants from multiple countries including China, USA, France, UK, New Zealand, the Netherlands, Brazil, Spain, and the Dominican Republic [21]; Italian students [22] and visitors to a Flemish agricultural fair [16] suggest fairly widespread acceptance of the environmental benefits of insect-based foods and feed as a driver for consumers’ acceptance of such products.

As well as environmental promotion, health promotion was found to be an important value for the acceptance of insect-based foods amongst respondents in Menozzi et al.’s [22] study, and this was supported by the findings of Lippi et al.’s [21] research.

When considering the health benefits of insect-based animal feed, Verbeke’s [16] study found that animals fed on insect-based diets were believed to be a significantly more difficult to sell compared to animals fed on conventional feed, but were believed to perform slightly better in terms of animal health. In addition, food products obtained from animals fed on an insect-based diet were perceived to have a higher nutritional value and to be healthier foods overall [16]. Pet health has been identified as a main driver of owners’ decisions in respect of their pets’ diets [54], with dog owners potentially found to be more serious about buying healthy dog food than healthy human food [55]. Valdés et al.’s [56] review of research into the health benefits (or otherwise) of insect-based pet foods found that the current body of research suggests insect nutrients, mainly amino acids, have high digestibility, are beneficial to health, do not have any detrimental effect on the gut microbiota and are accepted by dogs.

### 2.5. Attitudes towards Insect-Based Dog Foods

A person’s own willingness to eat insect-based foods has been found to be the main determinant of accepting the use of insects in animal feed [16]. This finding corroborates the view that food that is good for humans is also believed to be wholesome as feed for animals. Willingness to eat insect-based foods is also determined by a diversity of personal attitudes and interests (such as those set out above) as well as cultural exposure, familiarity or past experience and knowledge [16,57]. It is therefore anticipated that overall attitudes towards the benefits of insect consumption will influence attitudes towards IBDF and the intention to try IBDF.

### 2.6. Perceived Barriers

In addition to personal behavioural attitudes, further barriers to insect consumption and the purchase of IBDF may include factors such as price, availability and a lack of information [58].

Price is an important determinant for purchase intention, with the assumption that the higher the price, the lower the purchase intention will be. Herrmann et al. [59] noted that perceived price fairness has a significant positive influence on consumer satisfaction. As IBDF is still a relatively niche product in the UK, brands have not yet been able to take full advantage of the economies of scale that would allow IBDF to be offered at a lower price point than many nutritionally similar/equivalent dog-food products.

Information (or lack thereof) can also influence consumers’ attitudes and purchase choices towards plant-based diets [60]. Knowledge gaps in respect of products may mean that consumers are not aware of the benefits and/or risks, which may influence their attitudes towards the product or may even mean that the consumer is unaware of the existence of the product.

Availability (or lack thereof) is also important in influencing consumer attitudes and purchase behaviours in respect of particular products. It will not matter how positive a general attitude an individual has towards the consumption of insects and IBDF if the products are not available to purchase. A perception of the lack of availability will therefore keep purchase intention low and hinder purchasing behaviour [61]. Menozzi et al. [22] also identified a lack of products in the supermarket as a particular barrier to the eating of insect-based products.

### 2.7. Social Norms, Perceived Behavioural Control and Intentions to Try and to Buy

It has been predicted that the negative opinions of family members and friends may prevent Western consumers from eating insects in the future [25], and research has found that perceived insect-eating norms are a significant predictor of the intention to eat insect-based food products [62] and insect-tasting behaviour [29]. The findings suggest that perceived social norms play a substantial role in Western consumers’ willingness or lack thereof to eat insects [29]. Menozzi et al.’s [22] study also identified incompatibility with local food culture as a significant barrier to eating insect-based products.

A challenge in the introduction of novel food products to the market is having people try it as, once familiarity rises, purchase intention is known to rise with it [63]. Providing an option to try a product prior to purchase is often used in promotional activity because it offers increased consumer familiarity and helps build trust in novel products with little cost to the consumer [64].

Additionally, beliefs regarding factors which may hinder or facilitate the performance of the behaviour, and the strength of such beliefs, may determine perceived behavioural control [65]. Belief in the level of control over the behaviour expressed by individuals can change the behavioural outcome and influence consumer purchase intention [66]. Menozzi et al.’s [22] study found that attitudes and perceived behavioural control were statistically significant predictors of intention, and intentions and perceived behavioural control were significant predictors of behaviour in respect of eating insect-based products.

### 2.8. Conceptual Model

Based on the literature review, consumer attitudes towards IBDF are likely to be influenced by a number of factors, and previous research has revealed the complexity of European attitudes towards insects as food and feed [3]. A conceptual model based on the review of the literature is proposed, including the following hypothesised relationships based on the findings of the literature review:

**H1.** 
*Higher food neophobia has a significant negative effect on consumers’ attitudes towards insect-based dog food.*


**H2.** 
*Higher food neophobia has a significant negative effect on consumers’ intention to try insect-based dog food.*


**H3.** 
*Higher food neophobia has a significant positive effect on consumers’ feelings of disgust towards consuming insects and vice versa.*


**H4.** 
*Higher consumer feelings of disgust towards consuming insects has a significant negative effect on their attitudes towards insect-based dog food.*


**H5.** 
*Higher consumer feelings of disgust towards consuming insects has a significant negative effect on their intention to try insect-based dog food.*


**H6.** 
*Higher food neophobia has a significant positive effect on consumers’ perceptions of the risks associated with consuming insects and vice versa.*


**H7.** 
*Higher consumer perceptions of risks associated with consuming insects has a significant negative effect on their attitudes towards insect-based dog food.*


**H8.** 
*Higher consumer perceptions of the risks associated with consuming insects has a significant positive effect on consumers’ feelings of disgust towards consuming insects and vice versa.*


**H9.** 
*Higher consumer perceptions of risks associated with consuming insects has a significant negative effect on consumers’ intention to try insect-based dog food.*


**H10.** 
*Higher consumer perceptions of the benefits of insect consumption has a significant positive effect on consumers’ attitudes towards insect-based dog food.*


**H11.** 
*Higher consumer perceptions of the benefits of insect consumption has a significant positive effect on their intention to try insect-based dog food.*


**H12.** 
*Consumers’ attitude towards the use of animals has a significant effect on their attitudes towards insect-based dog food.*


**H13.** 
*Consumers’ attitude towards the use of animals has a significant effect on their preference for high animal welfare in food production.*


**H14.** 
*Consumers’ attitude towards the use of animals has a significant effect on their beliefs regarding insect sentience and vice versa.*


**H15.** 
*Consumers’ beliefs regarding insect sentience has an effect on consumers’ perceptions of the benefits of consuming insects.*


**H16.** 
*Consumers’ beliefs regarding insect sentience has an effect on consumers’ attitudes towards insect-based dog food.*


**H17.** 
*A higher preference for animal welfare in food production has a significant positive effect on consumers’ perceptions of the benefits of consuming insects.*


**H18.** 
*A higher preference for environmentally friendly foods has a significant positive effect on consumers’ perceptions of the benefits of consuming insects.*


**H19.** 
*A higher preference for healthy foods has a significant positive effect on consumers’ perceptions of benefits of consuming insects.*


**H20.** 
*Consumers’ attitude towards insect-based dog food has a significant effect on their intention to try insect-based dog food.*


**H21.** 
*A stronger consumer perception of barriers to insect-based dog food has a significant negative effect on their attitude towards insect-based dog food.*


**H22.** 
*A stronger consumer perception of barriers to insect-based dog food has a significant negative effect on their intention to buy insect-based dog food.*


**H23.** 
*Consumers’ social norms have a significant positive effect on their intention to try insect-based dog food.*


**H24.** 
*Consumers’ intention to try insect-based dog food has a significant effect on their intention to buy insect-based dog food.*


**H25.** 
*Consumers’ perception of behaviour control has a significant positive effect on their intention to buy insect-based dog food.*


The conceptual model is presented in Figure 1 below.

## 3. Materials and Methods

### 3.1. Data Collection and Sample

Ethical approval of the present study was granted by the University of Edinburgh’s Human-subject Ethical Review Committee. Data was collected via an anonymous online survey between 5 December 2022 and 28 February 2023. The survey was distributed using social media such as Facebook, LinkedIn and Instagram and on online UK pet forums. A brief introduction to the survey described the aims of the research as follows: “to investigate consumer acceptance among UK dog owners of insect-based dog food and the factors influencing consumer choice to try and buy such products.” Participants were asked to confirm their consent to participate in the study and were informed of the estimated time required to complete the survey (five–eight minutes) prior to completing the survey. Three initial questions were also asked to ensure that respondents were aged 18 or over, residents in the UK and owned or were responsible for the care of one or more dogs.

A total of 291 respondents completed the survey. However, 11 respondents did not meet the eligibility criteria so their responses were removed, reducing the total number of eligible respondents to 280. Previous studies have indicated that a sample size of 200 or above offers adequate statistical strength for structural equation model analysis [67], and the present dataset exceeds this minimum sample size. There was an observable skew towards female respondents (76.8%) and younger age groups (76.4% of respondents aged 45 or under). This is in line with other studies conducted using online platforms for data-collection purposes [68]. The proportion of vegan, vegetarian and flexitarian respondents were also over-represented compared to the UK population with 6.4% vegan respondents (compared to 3% of the UK population [69]), 10.7% vegetarian respondents (compared to 5–7% of the UK population [69]) and 26.1% flexitarian respondents (compared to 14% of the UK population [69]). A summary of the demographic characteristics of the respondents is presented in Table 1.

### 3.2. Survey Development

The survey was written and conducted in English as the most-spoken language in the UK. The survey included sets of questions derived from previous consumer-acceptance studies (a copy of which is available in the Appendix A) and was tailored to avoid repetition and to refer to canine nutrition where appropriate.

The first part of the survey included general questions designed to assess the attitudes of participants in the following areas: food neophobia; uses of animals; food preferences towards animal welfare, health and the environment; disgust; insect sentience; and the benefits and risks of insect consumption.

At the start of the second part of the survey, participants were provided with the following brief definition and description of IBDF: “Insect-based dog food is an innovative dog food with insects as the core ingredient. Seven species of insect are currently authorized for use in pet food in the EU and the UK of which the most commonly used are: house crickets, yellow mealworms and black soldier flies. Companies have already successfully launched insect-based dog food products in stores and online”. The second part of the survey then focussed on questions related specifically to IBDF. Participants were asked to answer questions related to attitudes towards IBDF; perceived behavioural control; perceived barriers; social norms; and intentions to try and buy IBDF.

All questions except those relating to eligibility and demographics were presented on a five-point Likert agreement scale (1 = strongly disagree, 5 = strongly agree). Data collected from the online survey were stored and managed using Microsoft Excel 2019 MSO (Version 2308 Build 16.0.16731.20052) and further processed by R version 4.1.0 using Lavaan and SemPlot package for data analysis [70]. All reverse-scaled questions were considered and reverse-scored when interpreted in the analysis. Table 2 shows the questions for all Survey Factors and Measurement Items and the references used to formulate these questions.

### 3.3. Data Analysis

The present study uses structural equation modelling (SEM) to test the causal links specified in the theoretical conceptual model presented at Figure 1. This is a two-stage process following Jöreskoget al. [78] and involving (i) a confirmatory factor analysis (CFA) to associate latent variables with their designed indicators and (ii) structural modelling to investigate the relationships between latent variables. The following three equations are used to specify the model:(1)x=Λxξ+δ,
(2)y=Λyη+ε,
(3)η=Bη+Γξ+ζ

Equations (1) and (2) relate to the CFA. Equation (1) relates observed indicators with the endogenous latent variables, where

x is a q × 1 vector of observed exogenous or independent variables;

Λx is a q × nmatrix of coefficients of the regression of x on ξ;

ξ is an n × 1 random vector of latent independent variables;

δ is a q × 1 vector of error terms in x.

Equation (2) relates observed indicators with the exogenous constructs, where

y is a p × 1 vector of observed endogenous or dependent variables;

Λy is a p × m matrix of coefficients of the regression of y on η;

η is an m × 1 random vector of latent dependent variables;

ε is a p × 1 vector of measurement errors in y.

Equation (3) defines the structural model and specifies the causal relationships which exist among the latent variables, describes the causal effects and assigns variances (both explained and unexplained). In equation (3),

B is a m × m matrix of coefficients of the η variables in the structural relationship;

Γ is a m × n matrix of coefficients of the ξ—variables in the structural relationship; and ζ is a vector of errors.

The variables were ordered, and not all variables were normally distributed, so the present study used the Diagonally Weighted Least-Squares method instead of Maximum Likelihood (ML) [79] because ML does not allow for the employment of the weight matrix required for the analysis.

The final stage of the process was to assess the goodness-of-fit of the model by analysing the factor loadings which relate each indicator with the constructs. The composite reliability and the average validity extracted for each construct was also measured [80]. Regarding the structural model, an assessment of the significance of the estimated parameters in the structural equations was carried out [80]. Finally, parameters such as Root Mean Square Error Approximation (RMSEA); goodness-of -fit index (GFI); the adjusted goodness-of-fit index (AGFI); the Comparative-Fit index (CFI); the Tucker–Lewis index (TLI); the Standardised Root Mean Square Residual (SRMR) and the Non-Normed-Fit index (NNFI) were also considered as indicators of the goodness-of-fit for the CFA and the SEM model.

## 4. Results

### 4.1. Descriptive Results

Figure 2 displays the percentage distribution of the agreement scale responses for each construct. For a more detailed review of the percentage distribution for each question within the constructs and the mean values for each construct, please refer to the Appendix A.

Eight constructs had an agreement percentage (both agree and strongly agree) of over 50% with food preference for health, the highest at over 85%. Perception of risks had the lowest agreement percentage though it is notable that over 55% of responses in relation to this construct indicated “Neither Agree Nor Disagree”, which suggests a level of uncertainty in respect of risks. In comparison, perceptions of benefits had agreement levels of almost 60%, and 36% of responses for this contract indicated “Neither Agree Nor Disagree”, which suggests greater levels of certainty in respect of the perceived benefits of IBDF.

### 4.2. Confirmatory Factor Analysis

During the initial CFA, items with factor loading < 0.5 were removed. Other variables were removed where the model would not converge with them. Namely, the latent variables of food neophobia, perceived barriers and perceived behavioural control were removed and the hypotheses associated with these variables (1, 2, 3, 6, 21, 22 and 25) were not included in the refined model. The indicators AUA4, AUA5, FPH3, AICB1 and AICB2 were also removed.

Further hypotheses were not included in the refined model as the model did not converge in respect of certain relationships. Of the original hypotheses H7, H20, H23 and H24 were supported and H8, H10 and H14 were supported in one path-direction only. The refined conceptual model with hypotheses is presented in Figure 3 below.

The construct and validity results of the estimated model are set out in Table 3. As presented, factor loadings varied between 0.577 for AUA2 to 0.987 for SN2. The Average Variance Extracted (AVE) ranged between 0.535 for AUA and 0.875 for SN. All factor loadings and AVE were above 0.5, which is the minimum acceptable figure indicated by Hair et al. [80]. The Alpha Ordinal varied from 0.761 for AUA to 0.946 for SN and all were above [81] a minimum of 0.7.

The goodness-of-fit was measured by the following indices: CFI and TLI both equal to 0.95; a RMSEA of 0.068; and a SRMR of 0.071.

### 4.3. Structural Model

An evaluation of the goodness of fit in respect of the refined hypothesised model showed that the model was a good fit for the data. The goodness-of-fit indices results were as follows: CFI = 0.95; TLI = 0.95; RMSEA = 0.065; SRMR = 0.078; GFI = 0.981; AGFI = 0.975; and NNFI = 0.985, which were all within the desired limits. The model can therefore be concluded to be an acceptable fit for the data.

The path diagram for the estimated SEM model is presented in Figure 4. All hypothesised paths were significant. The majority of the relationships identified were positive except for the following: (i) stronger perceptions of benefits relating toinsect consumption had a significant negative effect on the perception of risks of insect consumption, supporting H4 with a path coefficient of −0.377; (ii) stronger perceptions of risks relating to insect consumption had a significant negative effect on attitudes towards insect-based dog food, supporting H2 with a path coefficient of −0.304; and (iii) positive beliefs regarding insect sentience had a significant negative effect on the intention to try insect-based dog food, supporting H11 with a path coefficient of −0.325.

Higher levels of disgust were found to have a significant positive effect on perceptions of risks relating to insect consumption, supporting H1 with a path coefficient of 0.357.

In respect of attitudes towards insect-based dog foods, all pathways were significant at the *p* = 0.001 level, with the exception of food preference for health, which was significant at the 0.01 level. All pathways were positive with the exception of perception of risks relating to insect consumption. The perception of benefits of insect consumption had the highest influence on attitudes towards insect-based dog food, with a path coefficient of 0.543, followed by social norms with a path coefficient of 0.305 and food preference for health with a path coefficient of 0.179. Hypotheses 2, 3 and 5 are therefore supported. The level of positive influence of social norms on attitudes towards insect-based dog food is similar to the level of negative influence of perceptions of risks on attitudes towards insect-based dog foods.

In respect of influences on the intention to try insect-based dog food, all pathways were again significant at the *p* = 0.001 level, with the exception of food preference for animal welfare, which was significant at the *p* = 0.01 level. All were positive with the exception of beliefs regarding insect sentience. Social norms had the highest level of influence on the intention to try insect-based dog food, with a path coefficient of 0.568, followed by attitudes towards insect-based dog food with a path coefficient of 0.357. The negative influence of beliefs regarding insect sentience on the intention to try insect-based dog food is slightly lower than the positive influence of attitudes towards insect-based dog foods but higher than the level of positive influence of preference for animal welfare, which has a path coefficient of 0.280. Attitudes towards uses of animals has a significant positive effect on beliefs regarding insect sentience, with a path coefficient of 0.583. Hypotheses 7, 8, 9 and 10 are therefore supported.

In respect of the influences on intention to buy insect-based dog food, both pathways were positive and significant at the *p* = 0.001 level. The intention to try insect-based dog food had by far the most influence on the intention to buy insect-based dog food, with a path coefficient of 0.855. Food preference for environmentally friendly foods has lesser influence, with a path coefficient of 0.152. Hypotheses 12 and 13 are therefore supported.

The model also supported the indirect relationships set out in Table 4 below. The strongest indirect relationship identified was a positive influence of social norms on the intention to buy insect-based dog food via the intention to try insect-based dog food, with a path coefficient of 0.455. This was followed by the positive influence of food preference for animal welfare on the intention to buy insect-based dog food via the intention to try insect-based dog food, with a path coefficient of 0.313.

## 5. Discussion

### 5.1. General Discussion

This study seeks to investigate the factors contributing to the acceptance of IBDF in the adult UK dog-owning population. The results of this study support the conclusions of previous research [29,62] by demonstrating that social norms have a significant influence on consumers’ attitudes towards IBDF and the intention to try such products. However, the present research indicates that the influence of perceived social norms also extends to consumer attitudes towards insects as feed for their dogs and their intentions to try feeding such products to their dogs.

Social norms have been consistently cited as a driver of consumers’ acceptance of insects and other alternative protein sources [56] in consumers’ own diets and trends towards the humanisation of companion animals and companion animals’ diets [56], and previous research [15] suggests that dog owners wish to make similar choices for their dogs as they would for themselves.

Consumers’ ethical values were also found to have a significant impact on their attitudes towards IBDF. Consumers with preferences for purchasing food products with positive animal welfare and environmentally friendly credentials were more likely to have a positive attitude towards IBDF and have a stronger intention to try and buy IBDF, supporting the findings of previous research [16,21,22]. Given that producers of IBDF products tend to heavily emphasise the environmentally friendly aspects of the products, it is interesting to note that, while still significant, the preference for environmentally friendly food products had the lowest level of influence of any of the constructs in the model. In respect of preference for animal-welfare-friendly food products, the results of the present study suggest that this preference has a significant positive effect on consumers’ acceptance of IBDF. It is also interesting to note that in respect of the statement “insect-based dog food is better for animal welfare compared to most ordinary dog foods”, 112 participants (40%) agreed or strongly agreed.

The findings of this study indicate that stronger levels of belief in insect sentience have a significant negative influence on consumers’ acceptance of IBDF and that beliefs in insect sentience are, in turn, significantly influenced by consumer attitudes towards the uses of animals. This is in contrast to previous findings [26,27], which indicated that perceptions of insect suffering and attitudes in respect of the morality of killing insects did not significantly predict the acceptance of insects as food. It is notable that over half of respondents agreed or strongly agreed with the statements “I think that insects are capable of feeling pain” (156 = 56%) and “I think that insects are capable of experiencing suffering” (164 = 58%). However, levels of agreement notably drop in respect of the statement “Insects have consciousness” (128 = 46%) and reduced even further in relation to the statement “Insects have rights” (84 = 30%). For these two statements, 101 (36%) and 116 (41%) respondents, respectively, indicated that they neither agreed nor disagreed with the statement. It therefore appears that the concepts of consciousness and rights are more complex and possibly contentious than those of pain and suffering when it comes to insects. The contrast between the overall positive influence of consumer preference for animal welfare and the overall negative influence of beliefs in insect sentience on consumers’ acceptance of IBDF highlights a disconnect between concern for animal welfare generally and concern for insects in particular. Concern for animal welfare does not necessarily translate into concern for insect welfare.

In respect of preference for foods which are healthy for dogs, this study supported the results of previous studies [21,22], finding that such a preference has a significant positive influence on consumer attitudes towards IBDF. Somewhat surprisingly and in contrast to previous research [55], the level of influence was the second lowest. Consumer uncertainty around the healthiness of insect-based dog food for dogs may contribute to these results. In respect of the statement “insect-based dog food is healthy”, 174 (62%) indicated that they neither agreed nor disagreed with the statement. In respect of the statement “Insect-based dog food is safe for dogs to eat” 47% indicated that they neither agreed nor disagreed. In addition, 195 (70%) agreed or strongly agreed with the statement “I would be willing to try feeding insect-based food to my dog if it were recommended by vets”, suggesting that health is of significant interest to consumers and may be a stronger driver of consumer acceptance if the health benefits of IBDF were proven to a level that trusted animal-health professionals were willing to recommend it over more traditional diets.

While the model did not converge to include the influence of perceived barriers to consumers’ acceptance of IBDF, it is interesting to note that in respect of the statement: “I have little access to information about insect-based dog food”, 193 respondents (69%) indicated agreement or strong agreement. If consumers do not feel that they have sufficient access to information in respect of IBDF to allow them to make an informed choice, it is less likely that they would choose to opt for such a novel diet for their dogs.

The model also did not converge to include food neophobia. However, the model did show that higher levels of consumer disgust in respect of insects were related to stronger perceptions of risks associated with insect consumption. This is in line with multiple previous studies [22,25,26] in respect of the relationship between disgust and Western consumers’ attitudes towards eating insects and also supports findings [15] that disgust is a significant predictor of consumer’s willingness to feed insects to their dogs.

This study’s results show that both consumers’ perceptions of benefits and risks in respect of the consumption of insects have significant influence on their attitudes towards IBDF. Both also had clear indirect influences on the attitudes to try and to buy IBDF. Consumer perceptions of benefits had a greater level of influence than perceptions of risks, and stronger perceptions of benefits were also shown to have a significant negative effect on consumers’ perceptions of risk.

Perceived behavioural control was not supported within the model. This study also found that intentions to try feeding IBDF to their dogs has a very strong influence on intentions to buy such products, supporting the concept that facilitating increases in trying behaviour may promote future purchases through familiarity.

### 5.2. Future Research and Limitations

Participants in this study were recruited via online self-selection; therefore, sampling bias is likely to present an issue when attempting to generalise the results to the target population. The study sample was skewed towards younger demographics and, while the gender split is similar to that in a large scale “census-style” survey of adult UK dog owners [82], it is not necessarily representative of the gender profile of the adult UK dog-owning population as a whole. Previous studies have also found that females are likely to have more empathetic attitudes towards animal welfare [36,83], particularly in countries like the UK where females are more empowered [84,85]. Females are also more likely to express greater environmental concern and perceived environmental responsibility [86] and to limit meat intake for environmental reasons [87]. This potentially links to the higher proportions of vegetarian, vegan and flexitarian participants than is representative of the UK population [69], with such groups also being more likely to have greater concern for animal welfare and the environment [88]. However, a previous study has also found that vegans and vegetarians have been found to be over-represented among pet owners when compared to the general population [89].

This study did not collect demographic data in respect of the races and cultural backgrounds of respondents. The UK is a multicultural society and cultural biases are likely to have an influence on attitudes towards the consumption of insects as identified in the introduction to this paper. In addition, this study did not collect demographic information in respect of participant education levels or socio-economic groups. It is recommended that any replication of this study collects such data and, in addition, asks participants to confirm whether or not they feed or have previously fed IBDF to their dogs. For further research, we recommend collecting data either face-to-face or through a marketing research company and establishing quotas for gender, education, income, age and other relevant socioeconomic characteristics. Furthermore, SEM analysis employing multiple group analysis will allow researchers to understand better the behaviour of different consumer profiles.

Based on the findings of this research, food preferences for animal welfare, health and environmentally friendly production are all significant drivers for consumers’ acceptance of IBDF. It is therefore important to associate changes in canine diet towards insect-based products with animal welfare, health and environmental benefits. Where such benefits have yet to be proven or fully realised, it is recommended that producers, manufacturers, retailers and researchers work (together or separately, as appropriate) to ensure that robust research is carried out to assess the credentials of IBDF against these ethical parameters. In respect of animal welfare, research should be prioritised to better understand and monitor the welfare of insects in farm systems. It is also important to consider the health and welfare of the dogs, who may be the ultimate consumers of the food. Despite some evidence [56] that insect-based foods are beneficial to canine health, there is a significant research gap in respect of long-term health studies and comparisons to other diets. Further research in this area is also recommended in order that dog owners are able to make informed choices about what is best for their companions. These actions may also help to increase perceptions of benefits and reduce perceptions of risks in relation to IBDF.

Recent studies have identified general challenges [90] and species-specific insect welfare concerns for the insect-farming industry [91] and recommended more in-depth research into insect behaviour [92], species-specific needs, health, farming systems and humane methods of killing [93] in order to understand and promote insect welfare. Tackling ethical considerations in respect of edible insects and edible-insect production has been identified as a priority for further research [94,95,96,97]. There are also questions about the applicability to insects of animal-welfare frameworks and regulations developed with regard to vertebrate animals [96], so further research in this area would assist policymakers in producing suitable guidance and regulation in the industry in order to safeguard the welfare of all animals in the food chain.

The current marketing of IBDF tends to focus on potential sustainability and environmental benefits, but this study confirms that the motivations behind the acceptance of IBDF products are complex and multi-layered, with the preference for environmental products making up only a small part of the overall picture. Manufacturers and retailers should therefore consider diversifying the content of their marketing materials and packaging design to appeal to and reflect the complexity of consumer drivers for the purchasing of such products. Opportunities should also be offered to allow consumers to familiarise themselves with IBDF products.

## 6. Conclusions

Consumer acceptance of IBDF is multi-faceted, including social and cultural as well as ethical-preference components. Whilst the marketing of IBDF often leans into promotion of the environmental benefits of such products, it is clear that ethical concerns in relation to the environment make up only part of the puzzle in respect of UK dog owners’ acceptance of IBDF. It is important that consumers are able to make informed choices, particularly when their decisions have the potential to affect the health and welfare of other animals. Further research into the health and welfare effects of these products on insects themselves as well as the dogs who will be the ultimate consumers would inform law and policymakers in enabling the development of appropriate safeguards for all animals involved in the food chain and may give manufacturers and retailers scope to broaden marketing messaging.

In addition, it is clear that, even if a greater body of evidence were available to fill current knowledge gaps and inform choices based on consumer’s ethical preferences and concerns, this alone would likely have only a limited effect on consumers’ acceptance of IBDF among adult UK dog owners. In order to drive consumer acceptance further, action would also be necessary to address consumers’ perceived social norms and provide opportunities for consumers to build their familiarity with IBDF.

## Figures and Tables

**Figure 1 animals-14-01021-f001:**
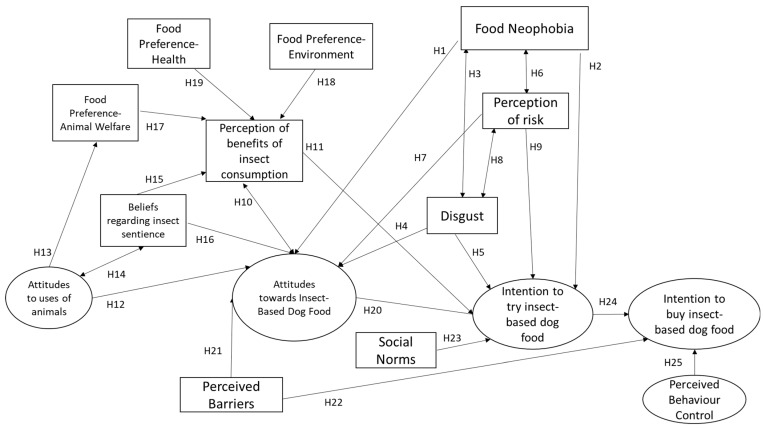
Conceptual model.

**Figure 2 animals-14-01021-f002:**
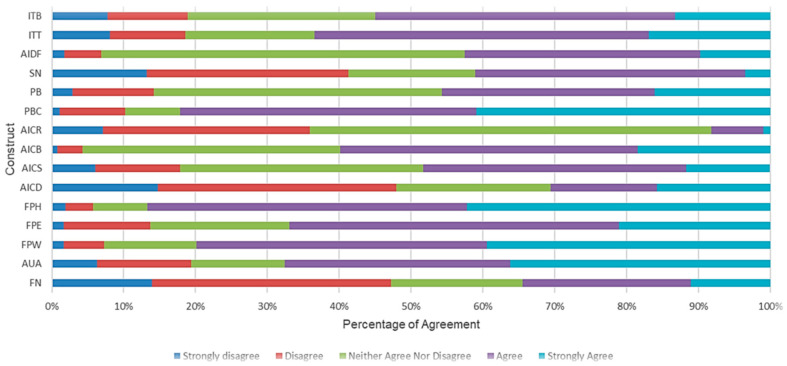
Agreement percentage distribution by construct.

**Figure 3 animals-14-01021-f003:**
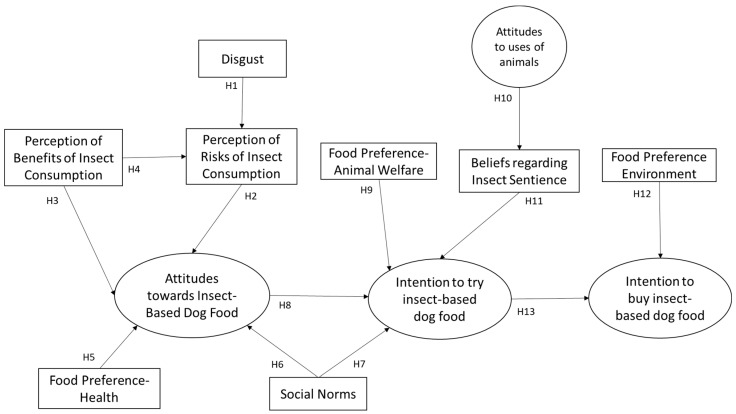
Refined conceptual model.

**Figure 4 animals-14-01021-f004:**
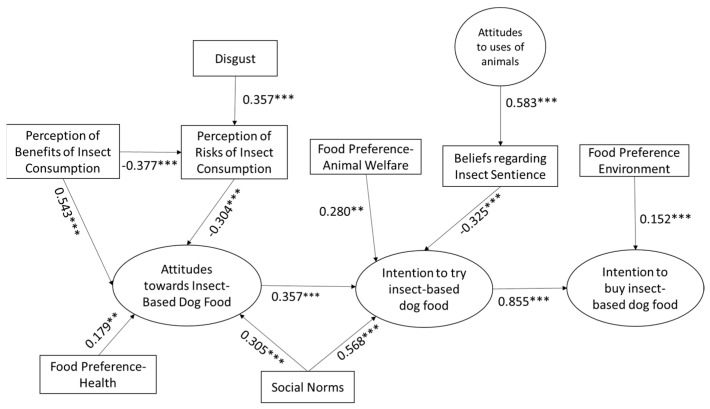
SEM path diagram. Key: ** statistically significant at 0.01 level; *** statistically significant at 0.001 level.

**Table 1 animals-14-01021-t001:** Sample demographics.

Variable	Categories	Number	Percentage
Gender	Male	63	22.5
	Female	215	76.8
	Other	2	0.7
Age	18–30	95	33.9
	31–45	119	42.5
	46–64	53	18.9
	65 and over	12	4.3
	Invalid response	1	0.4
Dietary Preference	Omnivore	157	56.1
	Flexitarian	73	26.1
	Vegetarian	30	10.7
	Vegan	18	6.4
	Invalid Response	2	0.7

**Table 2 animals-14-01021-t002:** Survey Factors and Measurement Items.

Factor and References	Measurement Items
Food Neophobia (FN) [17]	FN1 I am constantly sampling new and different foods *
	FN2 I do not trust new foods
	FN3 If I do not know what is in a food I will not eat it
	FN4 I am afraid to eat things I have never had before
	FN5 I will eat almost anything *
Attitudes towards Use of Animals (AUA) [33,34]	AUA1 It is morally wrong to hunt wild animals for sport
	AUA2 I do not think that there is anything wrong with using animals in medical research *
	AUA3 I think it is perfectly acceptable for cattle and pigs to be raised for human consumption *
	AUA4 The slaughter of whales and dolphins should be immediately stopped even if it means some people will be put out of work
	AUA5 I sometimes get upset when I see wild animals in cages at zoos
Food Preference Animal Welfare (FPW) [53,71]	FPW1 It is important to me that the food I buy is not normally produced by hurting animals
	FPW2 It is important to me that the food I buy has been produced in a way that respects animal rights
	FPW3 I think that more regulation is needed on how to treat animals in agriculture
Food Preference Environment (FPE) [53,72]	FPE1 My food purchasing habits are affected by my concern for the environment
	FPE2 I am worried about wasting the planet’s resources
	FPE3 I consider the potential environmental impact of my actions when I make many of my decisions
Food Preference- Health (FPH) [21,73]	FPH1 The healthiness of my dog’s food has little impact on my choice of food for them *
	FPH2 I am very particular about the healthiness of the food my dog eats
	FPH3 I feed my dog what he/she likes and do not worry much about the healthiness of the food *
	FPH4 It is important to me that the food my dog eats is nutritious
Disgust (AICD) [15,26,27]	AICD1 The idea of eating insects makes me feel nauseous
	AICD2 I am offended by the idea of eating insects
	AICD3 Eating insects is disgusting
Beliefs regarding Insect Sentience (AICS) [15,26,27]	AICS1 I think that insects are capable of feeling pain
	AICS2 I think that insects are capable of experiencing suffering
	AICS3 Insects have consciousness
	AICS4 Insects have rights
Perceptions of Benefits (AICB) [15,26,27]	AICB1 Rearing insects for food generates less pollution and greenhouse gas than rearing conventional livestock
	AICB2 Rearing insects as food is more efficient and requires fewer resources than rearing conventional livestock
	AICB3 Rearing insects for food requires much less space than rearing conventional livestock
	AICB4 Insects contain high levels of high-quality animal protein
	AICB5 Insects are highly nutritious
Perceptions of Risks (AICR) [15,26,27]	AICR1 Insects contain harmful toxins
	AICR2 Eating insects would expose me to harmful chemicals and insecticides
	AICR3 Eating insects will increase risk of infectious disease
Perceived Behavioural Control (PBC) [22]	PBC1 The decision to feed my dog products containing insect-based ingredients in the next month is under my complete control
	PBC2 Feeding products containing insect-based ingredients to my dog in the next month is completely up to me
Perceived Barriers (PB) [74]	PB1 Insect-based dog foods are more expensive
	PB2 Insect-based dog foods are not available in my local shops
	PB3 I have little access to information about insect-based dog foods
Social Norms (SN) [75]	SN1 I would be more likely to feed my dog insect-based dog foods if recommended by my family
	SN2 I would be more likely to feed my dog insect-based dog foods if recommended by my friends
	SN3 I would be more likely to feed my dog insect-based dog foods if recommended by my colleagues/peers
Attitudes towards Insect-based Dog Foods (AIDF) [21,68]	AIDF1 Insect-based dog food is healthy
	AIDF2 Insect-based dog food is safe for dogs to eat
	AIDF3 Insect-based dog food is more sustainable than most ordinary dog foods
	AIDF4 Insect-based dog food is better for animal welfare compared to most ordinary dog foods
Intention to Try Insect-based Dog Food (ITT) (Developed by authors)	ITT1 I would be willing to try feeding insect-based food to my dog if it were widely available in stores
	ITT2 I would be willing to try feeding insect-based food to my dog if it were recommended by vets
	ITT3 I would be willing to try feeding insect-based food to my dog if free samples were available
Intention to Buy Insect-based Dog Food (ITB) [15,76,77]	ITB1 I would buy insect-based dog food if it were produced in a more environmentally friendly way than ordinary dog food
	ITB2 I would buy insect-based dog food if it had more micronutrients than ordinary dog food
	ITB3 I would buy insect-based dog food if it were as accessible as ordinary dog food
	ITB4 I would buy insect-based dog food if it had a similar look and texture as ordinary dog food
	ITB5 I would buy insect-based dog food if it were from a renowned brand
	ITB6 I would buy insect-based dog food if it were cheaper than ordinary dog food

The asterisk indicates the measurement items which were reverse scored.

**Table 3 animals-14-01021-t003:** Factor loadings, Alpha Ordinal and AVE.

Factor and Item	Standardised Factor Loading	Alpha. Ordinal	Average Variance Extracted (AVE)
Attitudes towards insect-based dog food		0.889	0.696
AIDF1	0.859		
AIDF2	0.763		
AIDF3	0.853		
AIDF4	0.860		
Disgust		0.907	0.767
AICD1	0.818		
AICD2	0.888		
AICD3	0.918		
Perception of Risks		0.865	0.686
AICR1	0.772		
AICR2	0.826		
AICR3	0.884		
Attitudes towards uses of animals		0.761	0.535
AUA1	0.753		
AUA2	0.577		
AUA3	0.840		
Food preference: animal welfare		0.802	0.589
FPW1	0.696		
FPW2	0.784		
FPW3	0.818		
Attitudes towards insect sentience		0.903	0.740
AICS1	0.926		
AICS2	0.982		
AICS3	0.755		
AICS4	0.754		
Food preference: environment		0.919	0.815
FPE1	0.885		
FPE2	0.962		
FPE3	0.858		
Food preference: health		0.809	0.601
FPH1	0.703		
FPH2	0.758		
FPH4	0.857		
Social norms		0.946	0.875
SN1	0.958		
SN2	0.987		
SN3	0.856		
Intention to try		0.848	0.666
ITT1	0.904		
ITT2	0.726		
ITT3	0.809		
Intention to buy		0.924	0.697
ITB1	0.896		
ITB2	0.909		
ITB3	0.938		
ITB4	0.778		
ITB5	0.728		
ITB6	0.731		
Perception of benefits		0.797	0.681
AICB3	0.626		
AICB4	0.883		
AICB5	0.934		

**Table 4 animals-14-01021-t004:** Indirect relationships between constructs.

Indirect Relationship	Path Coefficient
AIDF → ITT → ITB	0.317
AICD → AICR → AIDF	−0.114
AICD → AICR → AIDF → ITT	−0.043
AICD → AICR → AIDF → ITT → ITB	−0.036
AICR → AIDF → ITT	−0.127
AICR → AIDF → ITT → ITB	−0.108
AUA → AICS → ITT	−0.253
AUA → AICS → ITT → ITB	−0.214
FPW → ITT → ITB	0.313
AICS → ITT → ITB	−0.268
FPH → AIDF → ITT	0.083
FPH → AIDF → ITT → ITB	0.070
SN → AIDF → ITT	0.103
SN → AIDF → ITT → ITB	0.087
SN → ITT → ITB	0.455
AICB → AICR → AIDF	0.160
AICB → AIDF → ITT	0.284
AICB → AICR → AIDF → ITT	0.060
AICB → AIDF → ITT → ITB	0.240
AICB → AICR → AIDF → ITT → ITB	0.051

## Data Availability

Authors will share the data collected upon request.

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
