# Peer review of "A Model for Consumer Acceptance of Insect-Based Dog Foods among Adult UK Dog Owners"

_animals, 2024, doi:10.3390/ani14071021_

Round 1

Reviewer 1 Report

Comments and Suggestions for Authors

Line 47 - 'production animals' - do you mean MEAT-production animals?

Line 54 - 'are overconsumed' - nothing is consumed until it is served. Dog caretakers are the only ones to blame for canine (and feline) obesity plague.

Line 95 (and throughout the whole manuscript) - [16] concluded... - it is an unusual form of literature citation. Why not Verbeke et al. [16]?

Reference 33 - why not using the original source https://doi.org/10.2752/089279391787057170 or newer paper by Harold Herzog, Stephanie Grayson & David McCord (2015) Brief Measures of the Animal Attitude Scale, Anthrozoös: A multidisciplinary journal of the interactions of people and animals, 28:1, 145-152?

Reviewer 2 Report

Comments and Suggestions for Authors

The title of this article effectively communicates the focus of the research, indicating that the study examines consumer acceptance of insect-based dog foods among adult UK dog owners. The abstract provides a concise overview of the study, highlighting the gap in research regarding consumer acceptance of insect-based dog foods and the methodology employed.

The introduction effectively highlights the relevance of insect-based feeds for companion animals, particularly dogs, considering sustainability concerns and changing consumer demands. To enhance the introduction, the authors could include a broader discussion of the implications of consumer acceptance of IBDF beyond the scope of this study, such as its potential impact on the pet food industry and sustainability efforts.

The literature review follows a logical structure, moving from general trends in alternative protein acceptance to specific considerations for IBDF, providing a clear flow of information. To enhance, it could include a synthesis of findings from previous studies to identify common themes and divergent viewpoints on consumer acceptance of IBDF.

The materials and methods are explained effectively. The study acknowledges that participants were recruited through online self-selection, which may potentially introduce a sampling bias. The article lacks comprehensive demographic data, particularly regarding race, cultural background, education levels, and socio-economic groups. This limits the ability to fully understand how these factors may influence consumer attitudes towards IBDF.

The discussion section effectively interprets the results of the study, discussing the implications of findings on consumer attitudes towards IBDF and highlighting the significance of key factors such as social norms and ethical values. To enhance the discussion, it could include a comparative analysis of the study findings with previous research to identify similarities, differences, and potential explanations for divergent results. To provide a more holistic discussion, it could include a section on the practical implications of the study findings for stakeholders in the pet food industry, policymakers, and consumers.

The future research and limitations section can be included in the later part of the discussion section. The section effectively identifies potential limitations of the study, such as sampling bias and the lack of comprehensive demographic data. It could be further enhanced by expanding on specific research questions or methodologies that future studies could explore in more detail. To further strengthen the section, it could include suggestions for mitigating the identified limitations in future research, such as alternative sampling methods or strategies for collecting more comprehensive demographic data.

Overall, given the increasing interest in alternative protein sources and sustainable pet food options, this study addresses a timely and relevant topic that has implications for both the pet food industry and broader sustainability efforts. Due to the same reason, there is a possibility of bias towards supporting the adoption of IBDF. Therefore, the article could benefit from a more balanced discussion of potential drawbacks and challenges associated with IBDF adoption. A brief discussion of practical implications of the study findings for stakeholders in the pet food industry, policymakers, and consumers would enhance the impact of this study further.
